# Using Wearable Sensors to Estimate Mechanical Power Output in Cyclical Sports Other than Cycling—A Review

**DOI:** 10.3390/s23010050

**Published:** 2022-12-21

**Authors:** Vera G. de Vette, DirkJan (H. E. J.) Veeger, Marit P. van Dijk

**Affiliations:** Department of Biomechanical Engineering, Delft University of Technology, 2628 CD Delft, The Netherlands

**Keywords:** cyclic sports, mechanical power, power output, wearable sensors, wearable technology, inertial measurement unit, IMU, power meter

## Abstract

More insight into in-field mechanical power in cyclical sports is useful for coaches, sport scientists, and athletes for various reasons. To estimate in-field mechanical power, the use of wearable sensors can be a convenient solution. However, as many model options and approaches for mechanical power estimation using wearable sensors exist, and the optimal combination differs between sports and depends on the intended aim, determining the best setup for a given sport can be challenging. This review aims to provide an overview and discussion of the present methods to estimate in-field mechanical power in different cyclical sports. Overall, in-field mechanical power estimation can be complex, such that methods are often simplified to improve feasibility. For example, for some sports, power meters exist that use the main propulsive force for mechanical power estimation. Another non-invasive method usable for in-field mechanical power estimation is the use of inertial measurement units (IMUs). These wearable sensors can either be used as stand-alone approach or in combination with force sensors. However, every method has consequences for interpretation of power values. Based on the findings of this review, recommendations for mechanical power measurement and interpretation in kayaking, rowing, wheelchair propulsion, speed skating, and cross-country skiing are done.

## 1. Introduction

Mechanical power is a useful and objective variable to monitor in cyclical endurance sports for several purposes. First of all, it can be used as a performance measure since the average velocity, and therefore performance, largely depends on the mechanical power sustained for a given distance [1]. In addition, mechanical power includes environmental factors such as wind velocity, which makes it an objective measure to assess the external load of a training or competition [2,3]. Furthermore, mechanical power can be used for fitness and fatigue assessments [3], and consequently, for prevention of overtraining and training periodization [4]. Therefore, estimations of mechanical power may be of great value for coaches, sport scientists, and athletes.

Most of the applications of mechanical power require day-to-day monitoring of mechanical power in an ecological valid environment. Therefore, in-field mechanical power estimation might be favorable for coaches and athletes as opposed to laboratory-based mechanical power estimation. Accordingly, in-field estimation of mechanical power in cycling is well-integrated in various cycling power meters, which are widely used by coaches, sport scientists, and athletes [5,6,7]. In cycling, power meters are often used to gain insight in power profiling, training load, and performance assessments and for establishing training zones [7]. As these applications of power are successfully developed in both professional and recreational cycling, the use of cycling power meters is an inspiration to provide methods for mechanical power estimation in other cyclical sports as well. However, having access to a commercially available power meter or a properly defined method to estimate mechanical power is not as common in any given cyclical sport as it is in cycling.

To estimate mechanical power in any sport of interest, it is important to understand its principle. In physics, power is defined as the rate of transferring energy from or to an object (i.e., doing work) with respect to time. Power associated with a force is calculated as the scalar product of the force vector and the velocity vector of its point of application, or F⋅v. In sports, mechanical power can be defined as the power transferred by the athlete to the environment, which is the main focus of this review. Mechanical power in sports can be estimated by solving the power equation while treating the human body as a chain of a number of linked rigid bodies [1]. Van der Kruk et al. [8] defined this power equation based on five terms: joint power, kinetic power, gravitational power, environmental power, and frictional power [8], with the following relationship:(1)$$P_{j}=P_{k}+P_{f}\;–P_{g}\;–P_{e}$$
where $P_{j}$ is joint power, $P_{k}$ is kinetic power, $P_{f}$ is frictional power, $P_{g}$ is gravitational power, and $P_{e}$ is environmental power. Translated into words, an athlete generates power ($P_{j}$) to (partially) overcome power losses due to resistive forces ($P_{f}$, $P_{g}$ and $P_{e}$) resulting in velocity and acceleration of the athlete ($P_{k}$).

According to van Ingen Schenau and Cavanagh [1] and van der Kruk et al. [8], mechanical power generated by an athlete can either be estimated by estimating the joint power (left-hand-side of Equation (1)) or through the sum of the kinetic power and power losses due to resistive forces (right-hand-side of Equation (1)). Joint power is calculated as the sum of the scalar products of joint moments and angular velocity per joint $\left(\sum M_{j}\;\cdot \omega_{j}\right)$ using inverse dynamics [1]. The power associated with resistive forces is calculated through the scalar product of the force and the velocity of its point of application. Hence, the right-hand-side of Equation (1), considering every acting force as an external force, simplifies to $\sum \left(\frac{dE_{kin}}{dt}\right)-\sum F_{e}\cdot v_{e}-\sum M_{e}\cdot \omega_{e}$ [1]. Therefore, the equation in Equation (1) can be rewritten as:(2)$$\sum M_{j}\cdot \omega_{j}=\sum \frac{dE_{kin}}{dt}-\sum F_{e}\cdot v_{e}-\sum M_{e}\cdot \omega_{e}$$

Obtaining an estimation of the power transferred from the athlete to its environment (e.g., rower on oar or boat, athlete to the wheelchair) using Equation (2), however, can be a very laborious procedure due to the number of variables that have to be measured and processed. Therefore, to increase feasibility for mechanical power estimation, simplifications of the power equation (Equation (2)) are often made. Frequently used simplifications and their consequences are extensively discussed by van der Kruk et al. [8]. One of these simplifications is using a single-body model in which the athlete is treated as a point mass located at the center of mass (CoM). Another simplification is the neglection of parts of the power equation: for example, by only taking what is considered as the main propulsion force into account for mechanical power estimation.

Using the power equation (Equation (2)) with or without simplification to obtain an in-field estimation of the power transferred from the athlete to its environment, requires wearable devices or sensors, such as strain gauges for force measurement or inertial measurement units (IMU) for measuring body segment kinematics [5,9]. IMUs are small and lightweight sensors that typically consist of an accelerometer, gyroscope, and magnetometer, which measure linear acceleration, angular velocity, and local magnetic field, respectively. With these outputs, IMUs can be used to determine segment kinematics such as orientation and angular velocity [9]. IMUs can also be useful for estimation of external forces [9], such that they could be used as a standalone approach for in-field mechanical power estimations. As many options and combinations for power estimation exist, and the optimal solution differs between sports, a sport-specific method is needed.

To summarize, a lot of possibilities exist for estimating mechanical power during in-field cyclical sports. There are different methods ($P_{j}$ or $P_{k}+P_{resistive}$) and multiple simplifications that can be made. Many decisions have to be made to establish a power model for a sport of interest, which can be challenging for coaches or sport scientists. To date, no overview of the methods to estimate mechanical power in different cyclic sports exists. Therefore, the aim of this review is to evaluate the literature on chosen approaches for estimating mechanical power, including the methods, devices and assumptions. By providing an overview and discussion of the existing methods, this review intends to guide coaches and sport scientists to form a well-founded model for mechanical power estimation in line with their intended aim.

As several reviews discuss the application of power meters in cycling [5,6,7], cycling will be omitted from this review.

## 2. Method

### 2.1. Literature Search

For this search, Scopus and PubMed were used. The last search was performed in May 2022. The complete search consisted of three search strings. The first search string included the following terms: mechanical power OR external power OR power output OR mechanical energy expenditure OR joint power OR internal power OR work rate. The second search included the following terms: cyclic sport OR swim * OR wheelchair OR kano * OR cross-country skiing OR speed skating OR skating OR rowing OR kayak. The third string included: IMU OR inertial sens * OR inertial measurement unit OR wearable sens * OR 3D acceler * OR force sens * OR power meter OR wearable devices OR wearable tech*. The strings were then combined using the AND modifier.

### 2.2. Selection of Studies

After removing duplicates, this search resulted in 16 records (Figure 1). Titles and abstracts were read to inspect whether the record was suitable for the current review. Records were included if they were focused on the method of estimation of mechanical power or any of the terms that are essential to estimation of mechanical power output and used healthy participants, with healthy meaning within the scope of the sport-specific requirements.

Records were excluded if one of the following exclusion criteria were present: published before the year 2000, focusing on non-cyclic sports, proposing methods that include energy harvesting of human locomotion, and not written in English. Applying the inclusion and exclusion criteria resulted in 8 relevant studies. Reference lists and citations of the selected studies were inspected for additional relevant reports, resulting in a total of 17 studies. The number of published studies over the years is shown in Figure 2.

## 3. Overview

An overview of the literature that estimated mechanical power using wearable devices is provided in Table 1 and Table 2. To identify the acting forces on a rigid body, drawing a free body diagram of that rigid body is a useful tool. The column ‘rigid body definition’ clarifies the boundaries of the rigid body that is used and therefore designating the forces and torques to consider. The studies of interest mainly estimate mechanical power as output (see Table 1). However, some studies in Table 2 estimated only subparts of the power equation, such as the acceleration of the center of mass [10] or push-off force [11,12]. These subparts are useful for future estimation of mechanical power and need to be accurately estimated as a first step towards mechanical power estimation in, respectively, rowing, speed skating, and cross-country skiing. Whichever term is estimated is displayed in the ‘estimated term’ column. Force or torque from a source acting on an object is displayed as $F_{source,object}$ or $M_{source,object}$ in the column ‘force measurement’. The kinematics of an object relative to a reference frame such as the linear velocity or angular velocity is displayed as $v_{object/ref}$ or $\omega_{object/ref}$, respectively, in the column ‘kinematic measurement’. If known, the type of sensor used to measure the force and/or velocity component is given between brackets. If the type of sensor used to measure the force and/or velocity component is unknown, it is displayed as (-). In Table 1, if a measuring system is commercially available, the name of the system is given.

### 3.1. Model

The studies are divided in two main categories based on the used rigid body: transportation object as a rigid body (Section 3.1.1) and athlete as a rigid body (Section 3.1.2). In one case, a combination of the transportation object and athlete is used as a rigid body [13].

**Table 1 sensors-23-00050-t001:** Overview of studies that estimated mechanical power output by using a part of the transportation object (i.e., paddle, oar, or wheel) as a system.

Sport	Study	Rigid Body Definition	Force Measurement (Sensor Type)	Kinematic Measurement (Sensor Type)	Commercially Available (Name)
Kayaking	Hogan et al. [14] Macdermid and Fink [15]	Paddle	$F_{hand,paddle}$ (SG)	$a_{shaft/world}\;$$\mathrm{and}\;\omega_{shaft/world}$ (IMU)	Yes (Kayak Power Meter)
Rowing	Baudouin and Hawkins [16]	Oar	$F_{hand,oar}$ (SG)	$\phi_{oar/boat}$ (POT)	No
	Doyle et al. [17]	Oar	$F_{oar,oarlock}$ (2D load transducers)	$\phi_{oar/boat}$$\;(\mathrm{POT})\;\mathrm{and}\;a_{boat/world}$ (ACC)	No
	Holt et al. [18]	Oar	$\mathrm{PowerLine}:\;F_{oar,oarlock}\;(-)$ $\mathrm{EmPower}:\;F_{oar,oarlock}\;(-)$ $\;\mathrm{OarPowerMeter}:\;F_{hand,oar}$ (-)	$\mathrm{PowerLine}:\;\omega_{oar/boat}\;(-)$ $\;\mathrm{EmPower}:\;\phi_{oar/boat}\;(-)$ $\;\mathrm{OarPowerMeter}:\;\phi_{oar/boat}$ (-)	Yes (PowerLine, EmPower, OarPowerMeter)
Wheelchair propulsion	Conger et al. [19]	Wheel	$M_{F_{hand,rim}}($SG)	$\omega_{rearwheel/WC}$ (-)	Yes (PowerTap SL+ Track Hub)
	de Groot et al. [20]	Wheel	$M_{F_{hand,rim}}$ (-)	$\omega_{rearwheel/WC}$ (-)	Yes (OptiPush, SMARTWheel)
	de Klerk et al. [21] van der Scheer et al. [22]	Wheel	$M_{F_{hand,rim}}$ (-)	$\omega_{rearwheel/WC}$ (-)	Yes (OptiPush)
	Mason et al. [23]	Wheel	$M_{F_{hand,rim}}$ (-)	$\omega_{rearwheel/WC}$ (-)	Yes (SMARTWheel)

ACC = accelerometer, IMU = inertial measurement unit, POT = potentiometer, SG = strain gauge, WC = wheelchair.

#### 3.1.1. Transportation Object as the Rigid Body

Nine studies estimated mechanical power by multiplying what is considered as the main propulsion force or torque with the corresponding linear or angular velocity (see Table 1) [14,15,16,17,18,19,20,21,22,23].

For kayaking and rowing, the paddle or oar, respectively, were chosen as the rigid body (see Figure A1) [14,15,16,17,18]. These studies considered the propulsive force as the force of the hands on the paddle or oar perpendicular to the oar or paddle ($F_{hand,\;paddle}$ or $F_{hand,\;oar}$) and multiplied this with the corresponding linear velocity of the hand relative to the world ($v_{hand/world}$) to obtain mechanical power. In rowing, $F_{hand,\;oar}$ can directly be measured at the hand placement on the oar [16,18]. Alternatively, it can be derived from the normal force in the oarlock ($F_{oar,oarlock}$), combined with the inboard and outboard length of the oar (respectively, $l_{in}$ and $l_{out}$) [16,17]. By assuming that the blade is a stationary point and the oar mass is negligible, this results in the following relation:(3)$$F_{hands,oar}=F_{oarlock,oar}\cdot \frac{l_{out}}{l_{in}+l_{out}}$$

The linear hand velocity ($v_{hand/world})\;$was derived by multiplying the inboard length with either the change in angle of the oar relative to the oar pin on the boat ($\phi_{oar/boat})$ divided by the corresponding change in time or the angular velocity of the oar relative to the oar pin on the boat ($\omega_{oar/boat})\;$[16,17,18]. Holt et al. [18] did not specify how the state-measured variables were derived. Since the PowerLine (Peach Innovations, Cambridge, UK) and EmPower (Nielsen-Kellerman, Boothwyn, PA, USA) measure $F_{oarlock,oar}$ [18], it is most likely that $F_{hands,\;oar}$ is derived using Equation (3), whereas $v_{hand/world}$ was derived similar to [16,17]. For the OarPowerMeter (Weba Sport, Wien, Austria), $F_{hands,oar}$ was directly measured and $\omega_{oar/boat\;}$ is likely multiplied with inboard length to obtain linear hand velocity [18].

**Table 2 sensors-23-00050-t002:** Overview of studies that estimated mechanical power or another essential term by using the athlete as a rigid body. In some estimations, a single body (SB) is used.

Sport	Study	Rigid Body Definition	Estimated Term	Force Measurement (Sensor Type)	Kinematic Measurement (Sensor Type)
Rowing	Kleshnev [24]	Rower	PO	$F_{oar,oarlock}$ (instrumented gates) $F_{feet,footstretcher}$ (SG)	$v_{seat/boat}$$,v_{trunk/boat}$$,\;\phi_{oar/boat}$$\;(\mathrm{POT}),\;v_{boat/world}$$\;(\mathrm{other}),\;a_{boat/world}$ (ACC)
	Lintmeijer et al. [10]	Rower	$a_{CoM/boat}$	-	$a_{seg/world}$ * (IMU)
Speed skating	van der Kruk et al. [11]	Skater	$F_{GR,feet}$	$F_{GR,feet}$ (3D force sensors)	-
Cross-country skiing	Gloersen et al. [25]	Skier (SB)	PO	$F_{GR,ath}=\left(m_{tot}a_{CoM}-F_{g}-F_{d}-F_{f}\right)\cdot \frac{v}{\left|v\right|}$	$v_{CoM/world}$ (IMU)
	Ohtonen et al. [12]	Skier	$F_{GR,feet}$	$F_{GR,feet}$ (SG)	-
	Uddin et al. [26]	Skier (SB)	PO	-	$a_{seg/world}\;$** (IMU)
Wheelchair propulsion	Rietveld et al. [13]	Wheelchair + athlete (SB)	PO	$F_{drag,ath}=m_{tot}\;\cdot a_{WC/world}\;$ $\left(\mathrm{IMU}\right)$	$v_{WC/world}$ (IMU)

ACC = accelerometer, IMU = inertial measurement unit, POT = potentiometer, SG = strain gauge, PO = mechanical power, SB = single-body model, WC = wheelchair. * seg = pelvis, abdomen plus thorax, head, the left and right thighs, shanks, feet, upper arms and the forearms plus hand. ** seg = chest, upper and lower back, left and right wrists, left and right skate

In kayaking, the normal force of the hand on the paddle ($F_{hand,paddle})$ was measured at the hand placement on the oar [14,15]. The hand velocity was derived using shaft acceleration ($a_{shaft/world})$, angular velocity of the shaft ($\omega_{shaft/world})$, and the hand placements [14,15].

In wheelchair propulsion, the wheel is chosen as a rigid body and the main propulsive force is considered the force of the hands on the rim tangential to the rim ($F_{hand,rim}$), resulting in a torque around the rear wheel axis ($M_{F_{hand,rim}}$) [19,20,21,22,23]. This torque is then multiplied with the angular velocity of the rear wheel around the rear wheel axis ($\omega_{rearwheel/WC})\;$to obtain mechanical power.

#### 3.1.2. Athlete as the Rigid Body

Alternatively, six studies estimated power using all forces acting on the athlete as a single or multibody model (see Table 2) [10,11,12,24,25,26]. One study estimated mechanical power by considering the athlete and transportation object as a single rigid body (see Table 2) [13].

A multibody model is used by both Kleshnev [24] and Lintmeijer et al. [10] in rowing. Kleshnev [24] determined all forces acting on the rower and their corresponding velocities to estimate mechanical power generated by the athlete (see Figure A2). The force of the foot stretcher on the feet in the propulsive direction ($F_{feet,footstretcher}$) was measured in the foot stretcher and $F_{oar,hand}$ was derived as $F_{oar,oarlock}$. The velocity of the feet is equal to boat velocity ($v_{boat/world\;}$) and $v_{hands/world}$ is derived using $\phi_{oar/boat}$ [24].

Based on a preliminary study of Hofmijster et al. [27], Lintmeijer et al. [10] determined the acceleration of the CoM of the rower relative to the boat ($a_{CoM/boat}$) in anterior-posterior direction. Multiplying $a_{CoM/boat}$ with the mass of the rower and the velocity of the boat and adding this to the power generated by the hands on the oar, results in an alternative mechanical power estimation for the rower, that, according to the authors, does not neglect any force in accordance with Equation (2).

Gloersen et al. [25] and Uddin et al. [26] used a single-body model of the athlete to improve the feasibility of mechanical power estimation in cross-country skiing, which was imitated with roller ski skating (see Figure A3). To further simplify the approach, both studies only used kinematic data and estimated the resistive forces to estimate mechanical power. Gloersen et al. [25] estimated the propulsive force of the ground on the athlete in the skiing direction ($F_{GR,ath}$) as the total mass of the athlete multiplied by the acceleration of the CoM of the athlete ($m_{tot}a_{CoM}$) minus the sum of power associated with gravity ($F_{g}$), rolling resistance ($F_{f}$), and aerodynamic drag ($F_{d}$) (right-hand-side of Equation (2)). An air drag model and rolling resistance coefficients were used to estimate the corresponding forces. The propulsive force ($F_{GR,\;ath}$) was multiplied with the velocity of the CoM to obtain a mechanical power estimation. Uddin et al. [26] performed their experiments on a treadmill, eliminating air drag. Mechanical power was calculated as the sum of power against gravity and rolling resistance. After obtaining an estimation of mechanical power, Uddin et al. [26] used a Long Short-Term Memory neural network to estimate the mechanical power during in-field roller ski skating based on data of seven IMUs, treadmill incline, and velocity and body mass. The relative error of the user-dependent model was 3.5%, while the relative error of the user-independent model was 11.6%. Considering this, the user-independent model is less accurate in estimating mechanical power, but it might be useful to recreational skiers.

Two studies determined the push-off force (i.e., the force of the ground on the athlete; $F_{GR,athlete}$) exerted by the athlete in speed skating and cross-country skiing (see Figure A3) [11,12]. These push-off forces are essential for mechanical power estimation and can be used in combination with kinematics to obtain a mechanical power estimation.

Only one study used a combination of athlete and transportation object as a rigid body [13]. Rietveld et al. [13] modelled the wheelchair with the athlete as a single body located at a point on the wheelchair. They assumed that the total CoM of this body is located at the wheelchair, which is rather acceptable over a push cycle. Mean drag forces were estimated based on the deceleration of the wheelchair during the non-push phase of wheelchair propulsion. However, in the non-push phase, the deceleration of the wheelchair is not necessarily equal to the deceleration of the CoM of the athlete plus the wheelchair since the upper body moves relative to the wheelchair in this phase. To obtain mechanical power, the mean drag force obtained in the non-push phase was multiplied with the velocity of the wheelchair [13]. However, Rietveld et al. [13] concluded that this method for mechanical power estimation is not yet suitable in wheelchair sprinting, due to the relative CoM movement, which is not taken into account.

### 3.2. Sensors

The sensors that were used in the considered literature can be divided into two main categories. The first category involves sensors that are able to directly measure mechanical power (Section 3.2.1), such as power meters. The second category uses separate kinematic and/or force sensors to obtain mechanical power (Section 3.2.2).

#### 3.2.1. Direct Mechanical Power Measurement

Seven studies used commercially available systems that are able to directly provide mechanical power (see Table 1) [14,15,18,19,20,21,22,23].

For kayaking, the Kayak Power Meter (One Giant Leap, Nelson, New Zealand) is designed and is reported to be applicable to both flat-water slalom and sprint kayaking. This power meter is validated by comparing mechanical power to the velocity of the kayak relative to water cubed ($v_{kayak/water}^{3}$) or to the velocity of the kayak relative to land cubed ($v_{kayak/land}^{3}$) in flat water conditions [14,15].

For rowing, three power meters are commercially available: the PowerLine, the OarPowerMeter and the EmPower. Holt et al. [18] recommended to use the PowerLine for measurement of mean and stroke-to-stroke mechanical power in rowing because of the higher sensitivity compared to the other two power meters.

Lastly, for wheelchair propulsion, three systems are available that can be used for mechanical power estimation. The OptiPush (Max Mobility, LLC, Antioch, TN, USA) and SMARTWheel (Three Rivers Holdings, Mesa, AZ, USA) were specifically designed for wheelchair propulsion. These two systems are not designed for mechanical power estimation; however, they provide the variables allowing for mechanical power estimation. The third system used in wheelchair propulsion is the PowerTap SL+ Track Hub (Saris Cycling Group, Madison, WI, USA), which is a power meter originally designed for cycling but modified to fit on a wheelchair.

To date, based on the current literature search, no commercially available power meter for speed skating and cross-country skiing exist.

#### 3.2.2. Force and Kinematic Measurement

For force measurement, mostly strain gauges were used, as can be seen in Table 1 and Table 2 [12,14,15,16,24]. Van der Kruk et al. [11] used three-dimensional piezoresistive force sensors to measure push-off force in speed skating (see Table 2). Several studies did not specify whatever type of force-measuring sensor was used [18,20,21,22,23]. Others modelled force as a function of other known components, such as acceleration [13,24].

For kinematic measurements after the year 2010, mainly IMUs were used (see Table 1 and Table 2) [10,12,13,15,25,26]. Before the year 2010, mainly potentiometers, occasionally in combination with accelerometers, were used [16,17,24]. Nowadays, these studies can be performed by replacing the potentiometers, used with or without accelerometers, with IMUs. For the studies in Table 2 which are not using commercially available power meters but are only using IMUs [10,13,25,26], any IMU can be used.

## 4. Discussion

The aim of the present review was to provide an overview and discussion of the present methods to estimate in-field mechanical power in different cyclical sports. Based on the sixteen studies considered in this review, the differences and similarities of the used mechanical power estimation methods were identified for application in kayaking, rowing, wheelchair propulsion, speed skating and cross-country skiing. By providing an overview of the current possibilities in mechanical power estimation in cyclical sports, this paper can be used as a guideline for coaches, sport scientists, and those interested in making well-informed decisions for estimation and interpretation of mechanical power.

The most extensive approach to estimate mechanical power is the joint power method (left-hand-side of Equation (2)) as discussed by van Ingen Schenau and Cavanagh [1] and van der Kruk et al. [8]. As this approach involves analyzing the full-body kinematics, the obtained mechanical power can be used as a measure of mechanical energy expenditure. However, this joint power method is quite laborious, which causes it to be less practical for coaches. In addition, if the aim is to obtain an accurate measure for an energy expenditure, doing a full-body kinematic analysis may defeat its purpose. It is probably more convenient to obtain an energy expenditure measure by directly measuring oxygen uptake with a wearable respiratory gas analysis device (e.g., Cosmed K5). Moreover, measuring oxygen uptake is a more accurate parameter for energy expenditure than mechanical power. Alternatively, heart rate can be used to indirectly estimate the oxygen uptake. This is, however, not recommended as the accuracy is low. Therefore, if the aim is to obtain an accurate measure for energy expenditure, direct measurement of oxygen uptake might be more favorable than using the joint power method.

Every other method to estimate mechanical power can be considered as a simplification of the joint power method to improve feasibility, such as the often-used main propulsion method. This method revolves around using the main propulsion force or torque in the specific sport, such as the force of the hands on the oar, paddle, or wheel. The main propulsion method is widely used in cycling power meters, where the force of the feet on the paddles is considered as the main propulsion force [5,6,7]. By multiplying the main propulsion force or torque with the corresponding linear or angular velocity vector, one obtains the mechanical power responsible for most of the propulsion. The main propulsion method is therefore useful when mechanical power is obtained to gain insights about performance.

Simplifications, however, are mostly accompanied by assumptions and implications to consider. For example, in rowing and wheelchair propulsion, there is a CoM movement of the athlete relative to the transportation object causing possible mechanical power transfer of the athlete to the transportation object, which is not accounted for with the main propulsion method. A simple thought experiment can clarify this: consider an athlete in a wheelchair moving his upper body in a periodic manner without applying force to the push rims. By doing so, the wheelchair is also moved periodically in a direction opposed to the trunk [28]. Since the wheelchair is moving, there is power loss due to rolling resistance. However, since there is no mechanical power input from the hands on the push rims, there has to be another location where mechanical power is added to the wheelchair. This situation was also explained by Hofmijster et al. [27] for rowing, where it presents itself when a rower in a boat only moves its body relative to the boat. However, over a cycle, it may be that the influence of this power input towards propulsion and therefore towards performance has no net contribution. The influence of other forces should be examined per sport in order to assess whether the main propulsion force method is sufficient or that other kinematics or forces should be measured.

If a sport, therefore, involves a transportation object and the intended aim is to obtain a mechanical power as a performance measure, it is advised to determine it with the main propulsion force. If the sport involves a transportation object and the intended aim is to obtain a measure for energy expenditure, consider using a respiratory gas analysis device instead of determining mechanical power using the joint power method. If a sport does not involve a transportation object and a measure for performance is desired, consider whether a simplification of the athlete such as a single-body model is sufficient. By doing so, the power associated with relative segment movements is neglected. However, it can provide information about general performance. If a sport does not involve a transportation object and a measure for energy expenditure is desired, again consider using a respiratory gas analysis device instead of determining mechanical power using the joint power method. A schematic overview to assist practitioners in the selection of a suitable power measurement method given their intended aim and type of sport is given in Figure A4.

If the main propulsion method is considered suitable for the set purpose, some commercially available power meters for kayaking, rowing, and wheelchair propulsion can be used. Although power meters are ambulatory and need almost no post-processing, the suitability of a power meter differs between sports. For kayaking and rowing, power meters are lightweight and thus, there is no influence on moving the equipment + power meter. However, for wheelchair propulsion, using the OptiPush or SMARTWheel for estimating mechanical power implies adding a considerable extra mass (7–9 kg per wheelchair). Chenier et al. [29] designed an instrumented wheel for wheelchair racing, which also adds 5.6 kg to an already lightweight racing wheelchair (8–10 kg). As those instrumented wheels increase the total weight of the wheelchair with ~50–90%, they will influence wheelchair dynamics. On top of that, these instrumented wheels may not be robust to collisions, making them not suitable for wheelchair field sports. To summarize, although instrumented wheels may be of use for assessing wheelchair biomechanics, the power measurements may not be convenient for daily wheelchair sport situations.

If power meters are not available or not practical, the appropriate kinematics and forces can be measured by means of IMUs or strain gauges. The future perspectives on using IMUs for mechanical power estimation are especially promising. Moreover, in some cases, measuring forces might be redundant with the use of IMUs. For instance, in wheelchair propulsion, it might be possible to estimate rolling resistance by placing an IMU on the wheelchair and one on the trunk [30]. Force on the rear and caster wheels could be modelled as a function of trunk angle and in combination with coast down tests for rolling resistance coefficients, rolling resistance could be estimated. Consequently, mechanical power can be estimated with the power lost to rolling resistance in combination with the estimation of the kinetic power (right-hand-side of Equation (2)). Power lost to air drag is then neglected, which is acceptable for low-speed indoor wheelchair sports, such as wheelchair basketball. Another option with IMUs is to use machine learning to estimate forces. For example, Uddin et al. [26] already used a Long Short-Term Memory neural network to estimate the mechanical power during roller ski skating by only using IMUs. This could also be an option for similar sports, such as speed skating or roller blade skating. Although some improvement is needed to make the method of Uddin et al. [26] suitable to implement on elite sport levels, this shows that IMUs in combination with machine learning have the potential to estimate mechanical power.

Lastly, the cost-effectiveness of estimating in-field mechanical power is useful to take into account when choosing the most appropriate approach. For equipment that can be used by multiple athletes or teams, such as a rowing boat, acquiring expensive power meters is soon affordable. In addition, once installed, the devices can remain on the boat. On the contrary, for wheelchair sports of speed skating, equipment should be purchased and installed for each individual wheelchair or skate. As athletes commonly have their own personalized equipment, using instrumented equipment for all athletes of a team will be both money- and time-consuming. Therefore, for sports with individualized equipment, non-invasive and cheaper solutions such as IMUs may be more feasible.

Although this literature review discusses the theory and practical implications of different in-field power measurement methods across different sports, some limitations should be noted. First of all, as this review was based on reported power measurement methods in kayaking, rowing, wheelchair propulsion, speed skating, and cross-country skiing, the most prominent pitfalls of those sports were discussed. Pitfalls of other sports may exist as well. However, as the concepts discussed in this review can be used as a guideline for mechanical power estimation in any other cyclical sport of interest, for example, swimming, canoeing, or roller blade skating, the main pitfalls of any other sport can be reasoned based on this review as well. Second, running was not taken into account in the present review. As the main focus of this review was defined by the power transferred from the athlete to the environment, which is only a fraction of the total mechanical power produced in running [1], running was considered beyond the scope of this review.

In conclusion, the most appropriate method to obtain mechanical power in cyclical sports differs for sports with transportation object compared to sports without a transportation object, and depends on whether performance or energy expenditure is the main interest. On top of that, the availability of a power meter, financial incentives, and mass of measurement equipment may influence the choice of a specific approach. This review provides useful handles to choose the most appropriate power measurement method for a given aim and type of sport, and explains the biomechanical underpinnings behind the different methods. A schematic overview to assist in selecting the proper power estimation method is given in Appendix A. With these insights, coaches, sport scientists, or any other person interested in measuring mechanical power can make their own, well-founded choices for measuring in-field mechanical power in any sport of interest.

## Figures and Tables

**Figure 1 sensors-23-00050-f001:**
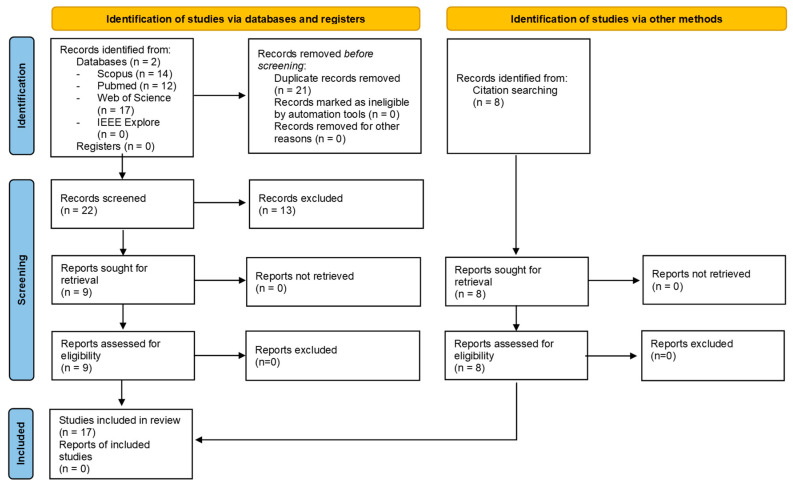
PRISMA diagram of included studies.

**Figure 2 sensors-23-00050-f002:**
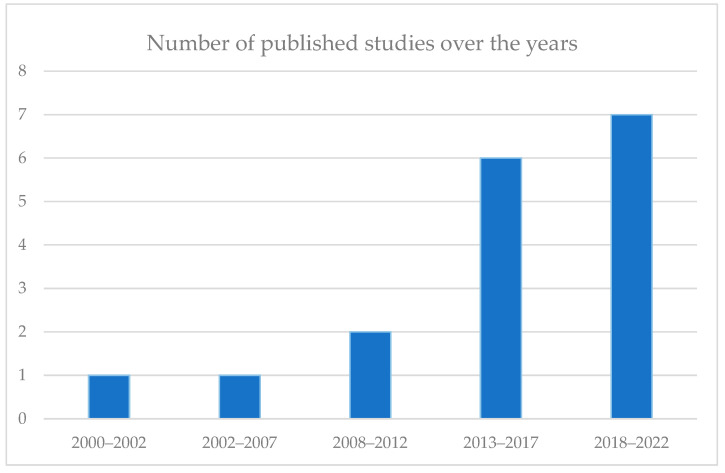
The number of published studies plotted against the year of publication.

## Data Availability

Not applicable.

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
