# Peer review of "Using Wearable Sensors to Estimate Mechanical Power Output in Cyclical Sports Other than Cycling—A Review"

_sensors, 2022, doi:10.3390/s23010050_

Round 1

Reviewer 1 Report

This review introduces the application of power sensing system in cyclical sports other than cycling. The manuscript can be considered for publication in Sensors after the authors address the following questions.

1.       To facilitate the author's elaboration and readers' understanding, some original schematic diagrams or authorized figures in the references should be considered to add in the manuscript.

2.       In lines 92 to 93 of this manuscript., it is explained the reason that the applications of power meter for cycling sports are omitted in this paper. But as far as I know, the cycling power meter is the most mature application. There are some relevant reviews introducing the application of power meter in cycling, but whether a concise introduction of the relevant content can be added in the manuscript. According to the successful application of power meter in cycling including meeting the needs of athletes, as well as the successful promotion of general consumers, provide inspiration for other sports.

3.       It is mentioned in the article that the monitoring of oxygen uptake is helpful to know the energy expenditure. However, the measurement devices of oxygen uptake can be divided into direct and indirect methods. The direct equipment is complex, mainly used in the laboratory environment. The indirect measurement is to calculate the oxygen uptake through the heart rate measurement and other methods, but the accuracy is poor. Before the large-scale application of power meters, heart rate meters were the main training equipment for athletes and coaches. Whether the article should consider more detailed introduction of the relevant content.

PowerTap SL+ Track Hub is mentioned in the section about sensors. The authors classify it as direct mechanical power measurement. However, the product introduced in Table 1 uses strain gauge as the sensor. This product is a commercial power meter. The working mechanism is to obtain the power value by multiplying the wheel shaft torque and the speed of the rear wheel. Whether there is ambiguity or overlap in this classification method, is there a more detailed or other classification.

Reviewer 2 Report

The presentation of this review paper is good and thorough. I have a few minor recommendations for the authors:

1) Authors only searched Scopus and PubMed databases. The references from 8 selected papers added 9 more papers: a total of 17, which may indicate that authors may require to look for other databases. I suggest authors double-check other databases.

2) Did the search only for papers written in English? Authors may mention it in the inclusion criteria.

3) As authors have included papers since 2000, it would be better if authors show the distribution of these 17 papers based on the year of publications.

4) Authors may provide more details on the sensors section, especially section 3.2.2 where they predicted IMU as a potential candidate for the next-generation infield power measurement tool. I suggest authors provide some theoretical overview of how the IMUs can be used. May be the working principle? Will all kinds of IMU work?

Reviewer 3 Report

1.       In the abstract, it is suggested to mention the cycling sports considered and the wearable sensors reported to monitor in-field mechanical power.

2.       After using the method explained on page 3, the authors have found 17 studies for the current review article. I am unsure whether 17 manuscripts are sufficient enough to prepare the review manuscript. All review articles need not be comprehensive. Still, 17 studies are too few to write a review article on any topic. I would appreciate the authors’ comments here.

 In my opinion, the title is misleading. There is nothing much literature on wearables sensors for in-field mechanical power estimation. Therefore, the motivation of the current article does not warrant the publication of the present review article.

Round 2

Reviewer 3 Report

The overall quality of the manuscript is improved.